# Unraveling the pathological biomineralization of monosodium urate crystals in gout patients
Carlos Rodriguez-Navarro [1] ✉, Kerstin Elert [1,2], Aurelia Ibañez-Velasco [1], Luis Monasterio-Guillot[1], Mariano Andres[3,4], Francisca Sivera[3,5], Eliseo Pascual[3,4] & Encarnación Ruiz-Agudo[1]

Crystallization of monosodium urate monohydrate (MSU) leads to painful gouty arthritis. Despite extensive research it is still unknown how this pathological biomineralization occurs, which hampers its prevention. Here we show how inflammatory MSU crystals form after a non-inflammatory amorphous precursor (AMSU) that nucleates heterogeneously on collagen fibrils from damaged articular cartilage of gout patients. This non-classical crystallization route imprints a nanogranular structure to biogenic acicular MSU crystals, which have smaller unit cell volume, lower microstrain, and higher crystallinity than synthetic MSU. These distinctive biosignatures are consistent with the template-promoted crystallization of biotic MSU crystals after AMSU at low supersaturation, and their slow growth over long periods of time (possibly years) in hyperuricemic gout patients. Our results help to better understand gout pathophysiology, underline the role of cartilage damage in promoting MSU crystallization, and suggest that there is a time-window to treat potential gouty patients before a critical amount of MSU has slowly formed as to trigger a gout flare.

Gout is a systemic illness associated with the deposition of monosodium urate monohydrate crystals (MSU, $NaC_5H_3N_4O_3 \cdot H_2O$) in articular and non-articular structures, typically synovial tissues and joint cartilage[1,2]. First described by Hippocrates (460-370 BC) and commonly known as the disease of Kings[3], gout prevalence and incidence have steadily increased over the last decades, with ~5 to ~23 million new cases per year worldwide[2]. MSU crystals induce painful inflammation (acute gouty arthritis or gout flares), as demonstrated by their injection in human and animal subjects[4,5]. They activate the complement system, myeloid cells (producing cytokines), neutrophils and NETosis, and the NLRP3 inflammasome[1,2,6,7]. In particular, MSU crystals trigger caspase-1-dependent prointerleukin (IL)-1β cleavage by activation of the NLRP3 inflammasome in macrophages and monocytes, which results in the release of active IL-1β[1,6,7]. Secretion of proinflammatory chemokines and cytokines results in the recruitment of neutrophils, increasing inflammatory cells infiltration and fostering a self-feeding necroinflammatory process[8]. Remarkably, gout flares spontaneously resolve after 7–14 days[2], but in absence of proper treatment can be recurrent or lead to chronic gouty arthritis, resulting in the development of gouty tophi, cartilage damage, and bone erosion[9,10]. Gout is typically associated with

hyperuricemia, that is, serum urate levels exceeding the solubility of MSU: 6.8 mg/dl (405 μM) at physiological conditions (pH ~7.4, ~150 mM [Na$^+$], and 37 °C)[2,11,12]. For reasons yet unknown, however, only <50% of hyperuricemic individuals develop gouty arthritis[12,13], which affects preferentially lower extremity weight-bearing joints where osteoarthritis (OA) with early cartilage damage is common[14]. This suggests that there must be other causative effects that trigger MSU deposition and inflammatory gouty arthritis, as it only affects specific body parts, despite similar levels of urate supersaturation throughout the body[15]. Among them, $T$, pH and ionic strength/specific ions (that affect urate solubility), damaged cartilage tissue[14,16,17], and/or different macromolecules have been considered[11,18–21]. Damage of connective tissue (e.g., cartilage) and secondary nucleation due to crystal shedding have been suggested to induce MSU crystal deposition[17,22]. OA and/or impact, or surgical tissue damage can also induce MSU formation and associated gout flares[2], assumed to occur following crystal shedding into joint space from MSU deposits present on and within surrounding cartilage and synovium[1,2]. Apparently, the formation of MSU and associated gout flares can be caused by a pathological condition(s) in addition to hyperuricemia. While much is currently known on the conditions that induce hyperuricemia and on the

[1]Department of Mineralogy and Petrology, University of Granada, Fuentenueva s/n, 18002 Granada, Spain. [2]Escuela de Estudios Arabes, Consejo Superior de Investigaciones Científicas (EEA-CSIC), C. Chapiz 22, 18010 Granada, Spain. [3]Department of Clinical Medicine, Miguel Hernandez University, CN 332 s/n, 03550 Alicante, Spain. [4]Department of Rheumatology, Dr. Balmis General University Hospital, Alicante Institute for Health and Biomedical Research, Av. Pintor Baeza s/n, 03010 Alicante, Spain. [5]Department of Rheumatology, Elda General University Hospital, Carretera Elda-Sax s/n, 03600 Elda, Spain. ✉e-mail: carlosrn@ugr.es

inflammatory cascade caused by MSU crystals[1], little is known about the in vivo formation of MSU crystals. This is a strong handicap for the prevention of gout and its treatment.

Specific (macro)molecules (or structures) in the synovial fluid (SF) or joint space could promote the (heterogeneous) nucleation of MSU[15,19,20,23]. Conversely (other) macromolecules might inhibit its formation[18]. The presence of the former or the lack of the latter could thus result in MSU precipitation in cases where supersaturation with respect to MSU in the SF exists[15]. However, for in vitro MSU to precipitate under physiological conditions, a very high critical supersaturation has to be reached, with urate levels at least one order of magnitude higher than those observed in hyperuricemic subjects[19]. It follows that for MSU to crystallize in vivo there must be a promoter rather than the lack of an inhibitor. This is consistent with observations of the promoting effect of SF from gouty patients on MSU crystallization[15,19]. Yet little is known regarding the macromolecule(s) and/ or structure(s) responsible for such a promoting effect, although it has been suggested that it involves a protein or protein structure since the promotion effect of SF of gouty patients is lost upon thermal treatment[15,23].

The SF includes high levels of hyaluronic acid (HA), along with other macromolecules such as serum proteins albumin and immunoglobulins[20]. There are contrasting results regarding the role of HA component on the crystallization of MSU: it has been reported that it does not seem to affect the crystallization of MSU[19], while recent studies indicate that it is a crystallization inhibitor[24,25] Conversely, the proteins can interact with MSU crystals and may promote their template-directed crystallization[20], even involving highly specific antigen-antibody interactions, as proposed for the case of IgG[26,27] and IgM antibodies[28]. The in vivo demonstration of MSU precipitation induced by antibodies in humans is, however, lacking. Moreover, this highly specific lock-and-key antibody-antigen (i.e., MSU) interaction could not trigger the nucleation of an amorphous MSU precursor phase (AMSU) with a disordered structure that does not match the structural imprint in an IgG (or IgM) antibody template. Note that AMSU, which interestingly is not phlogistic[5], has been observed to precede the formation of MSU in vitro[29,30]. There is, however, no evidence yet as to the formation of AMSU in vivo. It has also been shown that irrespective of an antibody-antigen effect, IgG promotes the nucleation of MSU in vitro[23]. However, immunoglobulins, IgG in particular, exist in relatively high concentrations in SF of both gouty and non-gouty individuals, which casts doubts on whether the observed promoting effect is enough to trigger MSU precipitation in vivo. The presence of damaged cartilage might also induce MSU (or AMSU) precipitation, favoring heterogeneous nucleation, which might also occur in tendons[17]. McGill and Dieppe[23] showed that Type I collagen (i.e., present in tendons) promoted MSU crystallization in vitro. Recent research has shown that human cartilage homogenates (i.e., made of dispersed collagen fibers) influence MSU crystallization (smaller crystals) and the activation of the inflammasome[31], and injured (i.e., fibrillated) Type II collagen (i.e., present in cartilage) upregulates MSU crystallization and inflammatory cell recruitment[8]. There is also evidence showing the association of MSU with cartilage degradation involving collagen defibrillation in humans[14]. Yet, how damaged cartilage might promote the precipitation of MSU is not known.

Interestingly, biomacromolecules (proteins and polysaccharides), both soluble and insoluble (organic matrix), involved in the non-pathological biomineralization of a range of biominerals (e.g., mollusk shells, sea-urchin spines, and bones) are known to control their nucleation and growth[32]. Typically, they affect the shape and size of crystals forming a biomineral structure and regulate their non-classical crystallization via interaction with amorphous precursors and final crystalline phases[33,34]. Organic macromolecules can also inhibit nucleation, both in vitro and in vivo, when present in the bulk solution, or promote nucleation when adsorbed (as a template) on a substrate or forming a matrix[32]. Ultimately, the biomineralization process imprints distinctive textural and structural features to biominerals that affect their physical-chemical properties and help disclose their formation mechanism[32-35]. It is however unknown if such biomineralization features are present in biotic MSU crystals.

Here we aim to disclose whether MSU of gouty patients forms non-classically via an amorphous precursor and displays specific features common to other biominerals, that can fingerprint its biogenic origin and unravel its formation mechanism. For this task we studied the nano- and microstructural features of both abiotic (synthesized in the laboratory) and biogenic MSU from gouty patients.

## Results

### Phase and morphology of biotic MSU from gouty patients

The presence of biogenic MSU precipitates in the SF extracted from gouty patients (see Methods) was confirmed by standard powder X-ray diffraction (XRD)[36], Fourier transform infrared spectroscopy (FTIR), Raman spectroscopy, and thermogravimetry coupled with differential scanning calorimetry (TG-DSC) (Fig. 1a–d and Supplementary Fig. S1). Polarized light microscopy (PLM) showed needle-shaped MSU crystals in all studied SF samples ($N = 5$), with their characteristic negative elongation under compensated crossed-polar observation (Fig. 1c)[16]. Under the field emission scanning electron microscope (FESEM) (Fig. 1d, e) and transmission electron microscope (TEM) (Fig. 1f–h, Supplementary Fig. S2), MSU crystals showed marked acicular habit (1–20 μm long, 0.5–1.5 μm thick) and displayed a nanogranular surface topography, as previously reported[37]. Such nanogranular features are similar to those observed in $CaCO_3$ biominerals and their biomimetics formed via a non-classical crystallization route involving an amorphous precursor and nanoparticle aggregation-based growth in the presence of biomacromolecules[34]. Unfortunately, MSU crystals underwent rapid and extensive beam damage under the TEM[37], which prevented high-resolution lattice imaging. In the case of a gouty patient with tophi, who showed inflammation due to abundant crystal shedding into the joint space, the SF displayed a milky appearance (i.e., so-called urate milk)[38] and the dispersed MSU crystals showed the same above-indicated structural and textural features of MSU crystals in SF. In another gouty patient with tophi, it was possible to extract with a surgical needle an intact fraction of the tophus, whose structure was formed by compact aggregates of MSU crystals oriented along [001] (Fig. S1a), with dimension and morphology similar to those of crystals dispersed in the SF of the other gouty patients studied. One difference, however, was the presence of abundant N-rich organics (i.e., proteins) surrounding the latter crystals (Fig. S2h–j) and their lack in the tophus structure. Detailed analysis by FESEM and TEM/selected area electron diffraction (SAED) in combination with high angle annular dark field (HAADF) imaging (Z contrast) and energy dispersive X-ray spectrometry (EDS) showed that AMSU was present in the SF aspirates along with MSU crystals (Fig. 2, Supplementary Figs. S3 and S4). AMSU nanoparticles of a few tens of nm in size were found attached to collagen fibers several μm long (clearly identified by their standard dark-light D-banding normal to the fiber axis) (Fig. 2a). The presence of this amorphous sodium urate phase was confirmed by the diffuse haloes in the SAED pattern (inset in Fig. 2a) and the Na and N EDS maps (Fig. 2a). These results show that AMSU is a precursor of biotic crystalline MSU and its formation is templated (heterogeneous nucleation) by collagen fibrils, likely derived from damaged cartilage[14]. In one synovial fluid sample, spherulitic beachball-like structures ~1 μm in diameter were detected. They displayed a nanogranular surface texture (Fig. 2b) and were made up of AMSU nanoparticles (Fig. 2c, d). Such beachball-like structures previously found in SF of gouty patients were, however, assumed to be made of crystalline MSU[39]. Similar spherulitic structures, also made up of crystalline MSU, have been synthesized in vitro[24,39,40]. Considering that AMSU is metastable and more soluble than MSU (see below), we hypothesize that these biogenic AMSU beachballs can act as a reservoir of sodium and urate ions for the subsequent crystallization of (abundant) MSU crystals. In other SF samples, both AMSU aggregates and MSU crystals (~1 μm long) were observed in close association, as shown by TEM-SAED (Fig. 2e–g) and HAADF-EDS analysis (Fig. 2h), which points to the formation of the latter after AMSU. Altogether, these textural and nano/microstructural features show that MSU of gouty patients forms via a non-classical crystallization and growth mechanism involving a metastable amorphous (AMSU)

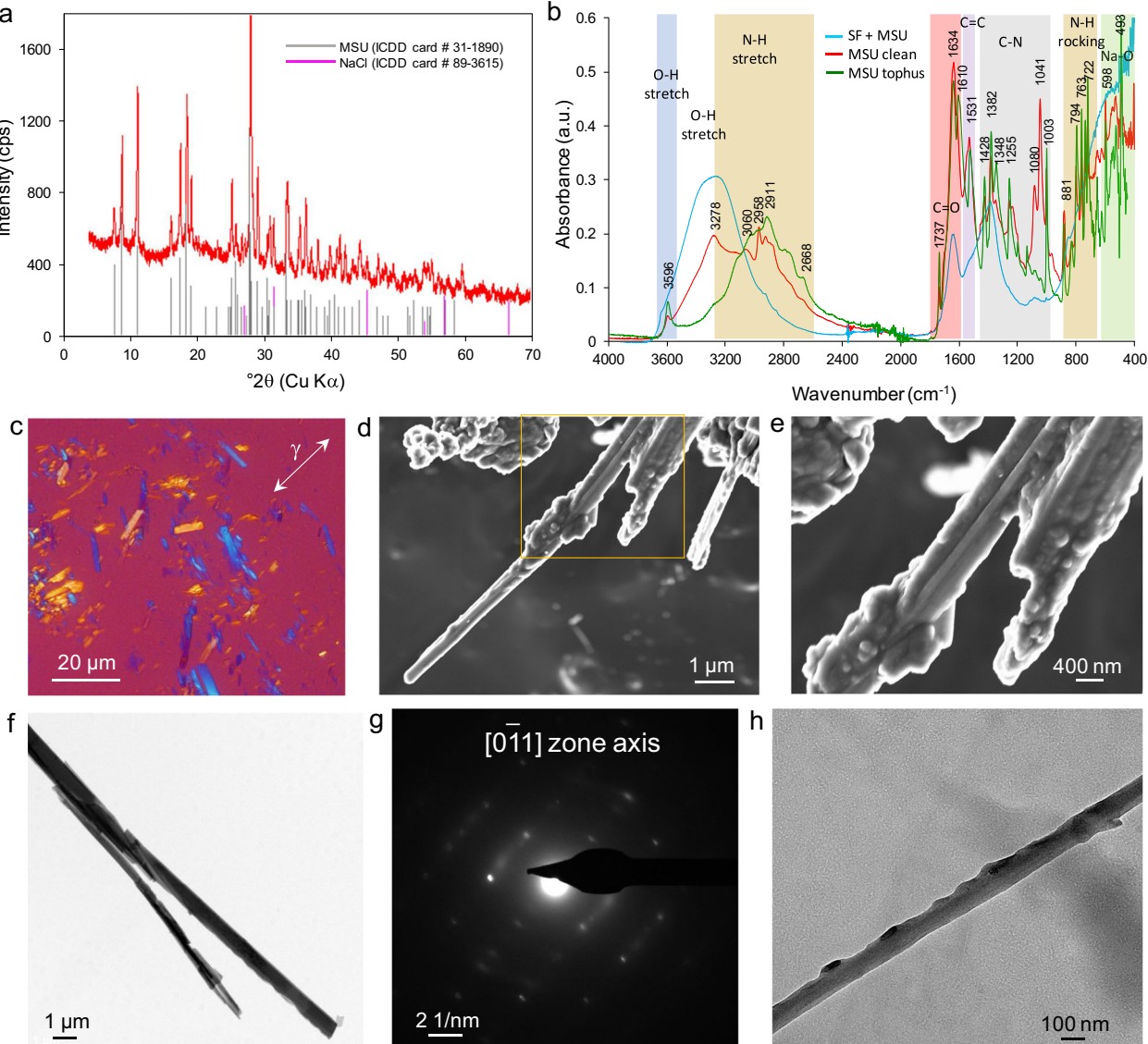

**Fig. 1 | Characterization of human MSU in synovial fluids. a** Representative XRD pattern of MSU (with small amounts of NaCl crystallized during drying of the samples: ICDD card number for the MSU and halite phases are indicated, as well as the position of their Bragg peaks); **b** FTIR spectra of biotic MSU before and after synovial fluid protein elimination (proteinase K treatment for the proper visualization of the main IR bands of MSU crystals). Main bands and corresponding molecular vibration are indicated (shaded areas); **c** Polarized light microcopy image (with 550 nm compensator) showing MSU crystals with their distinctive negative elongation; **d** FESEM image of MSU fibers, and **e** detail of the fiber in the squared area of (**d**) showing the nanogranular surface structure. **f** Representative TEM image of MSU crystals. **g** SAED pattern of the larger fiber in (**f**) demonstrating it is crystalline MSU. **h** Detail of a MSU fiber showing a nanogranular topography indicative of a non-classical growth via aggregation of nanoparticles. Nanogranular features are not caused by protein adsorption because the MSU crystals in this image were treated with proteinase K to eliminate any absorbed protein.

precursor[34]. Our results also show that the initial formation of AMSU is templated by collagen fibrils. Finally, it is revealed that the textural and nano/microstructural features of MSU crystals dispersed in SF are similar to those in tophi, pointing to a common origin for both.

## Phase and morphology of abiotic MSU

All in vitro crystallization routes explored here (i.e., crystallization in silica gel, crystallization by titration, and by rapid mixing of uric acid + NaOH and NaCl solutions at 37 °C, pH 7.4 ± 0.4; see Methods) resulted in the formation -within hours to few days- of MSU crystals as shown by XRD, TG/DSC and FTIR analyses (Supplementary Fig. S5a–c). Irrespective of the precipitation route, MSU crystals were needle-like, or more exactly, blade-like, elongated along [001], and commonly displayed {110} twining[41]. Note that we observed no twining in biotic MSU. Abiotic MSU crystals tended to be slightly larger than their biotic counterparts, with

length ranging from 2 up to 25 μm and width from 0.5 up to 2 μm as shown by FESEM and TEM observations (Supplementary Fig. S5d–g). Otherwise, biotic and abiotic MSU showed very similar textural and structural/compositional features, with the exception of gel-grown MSU crystals, which were surrounded by silica gel (Supplementary Fig. S6). Precipitation in titration experiments, detected by a drop in transmittance until a stable minimum was reached (Fig. 3a), occurred at 3275 ± 283 s (*N* = 4) following the addition of 3 M NaCl solution (at a dose rate of 0.125 ml/min) to the 13.3 mM uric acid buffer (100 ml, pH 7.4, 37 °C) (see Methods). Solids collected at the onset of precipitation were poorly crystalline, as shown by XRD analysis (Fig. 3b). They consisted of nanogranular/spherulitic aggregates made up of individual spheroidal nanoparticles ~20–60 nm in diameter (Fig. 3c, d). They corresponded to AMSU as demonstrated by the diffuse haloes in their SAED patterns (Fig. 3e). We observed coexistence of AMSU and MSU in these early

**Fig. 2 | Amorphous precursor of biotic MSU.**
**a** TEM image of a collagen fiber spotted with amorphous MSU (AMSU) precipitates on its surface. The corresponding Na and N EDS maps demonstrate that the precipitates are AMSU. **b** FESEM image of beachball-like AMSU precipitates. Their surface shows fibers (**c**) which are amorphous, as shown by the diffuse haloes in the SAED pattern (**d**). **e** TEM image (bright field) of an aggregate of AMSU and MSU. The corresponding SAED patterns show that the blue circled area is amorphous (**f**) and the red circled area is crystalline (**g**). **h** HAADF image of an aggregate including both AMSU and MSU. The Na and N compositional maps demonstrate that these two phases are monosodium urate.

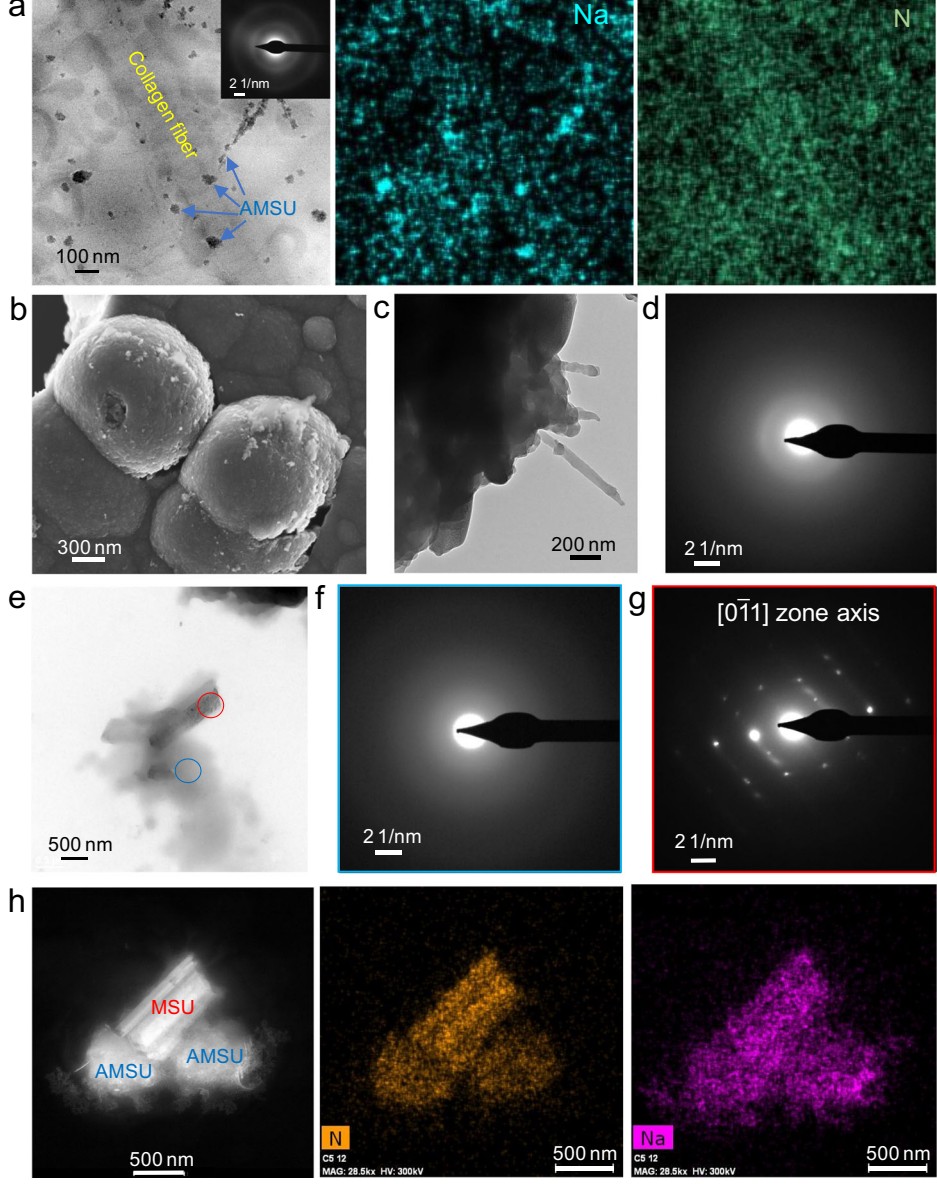

precipitates (Fig. 3f), which shows that the conversion of AMSU into MSU is rapid under our precipitation conditions. In all cases, EDS analyses showed the presence of C, O, N and Na (Fig. 3g), demonstrating that the precipitates (both amorphous and crystalline) were sodium urate phases. Precipitates collected at the end of the titration experiment were more crystalline (Fig. 3b) and made up of acicular MSU crystals (Fig. 3h–j). Interestingly, FESEM showed nanogranular aggregates making up the early formed MSU fibers (i.e., samples collected at the early stage of precipitation) as in biotic MSU (see above), which suggests that both biotic and abiotic MSU forms via aggregation of AMSU nanoparticles followed by their amorphous-to-crystalline transformation, as observed in biomimetic CaCO₃[42]. However, in contrast to biotic MSU, no nanogranular features were observed in the well-developed abiotic MSU crystals collected at the end of the precipitation tests. Note that in the case of CaCO₃ biominerals and their biomimetics, the preservation of the nanogranular structure following the conversion of amorphous calcium carbonate (ACC) into crystalline CaCO₃ (e.g., calcite) is associated with the presence of macromolecules[34]. It is thus likely that the absence of a nanogranular structure in the final abiotic crystalline MSU is due to the lack of macromolecules in the precipitation medium. Conversely, adsorbed macromolecules (e.g., proteins such as IgG)[1,23] in the case of biotic MSU crystals,

likely contributed to the preservation of the observed nanogranular surface structure after the AMSU-to-MSU conversion.

Our analysis of the time evolution of conductivity during titration experiments showed that the measured conductivity was lower than the calculated conductivity at any time point before precipitation (Fig. 3a) (see Methods for details regarding how the theoretical conductivity was calculated). This suggests that prior to the onset of AMSU precipitation there was ion binding between urate and sodium ions, likely forming polynuclear ion associates (i.e., prenucleation clusters, PNC) as observed in other systems[43,44]. These entities play a critical role in the non-classical nucleation of inorganic and organic solids as their aggregation and dehydration reduce cluster dynamics and result in the formation of dense liquid and/or solid amorphous precursors[43,44], in this case AMSU. Simple ion-pairing cannot account for the observed degree of ion binding (i.e., resulting in the deviation between calculated and measured conductivity), because the formation of neutral sodium-urate ion pairs has been shown to be negligible under physiological conditions[45]. It follows that larger ion associates must be present before precipitation. These results further show that MSU crystallization is non-classical. Conductivity measurements enabled the determination of sodium and urate activities and the saturation index, *SI* at the onset of precipitation [$SI = \log(IAP/K_{sp})$, where *IAP* is the ion activity product and

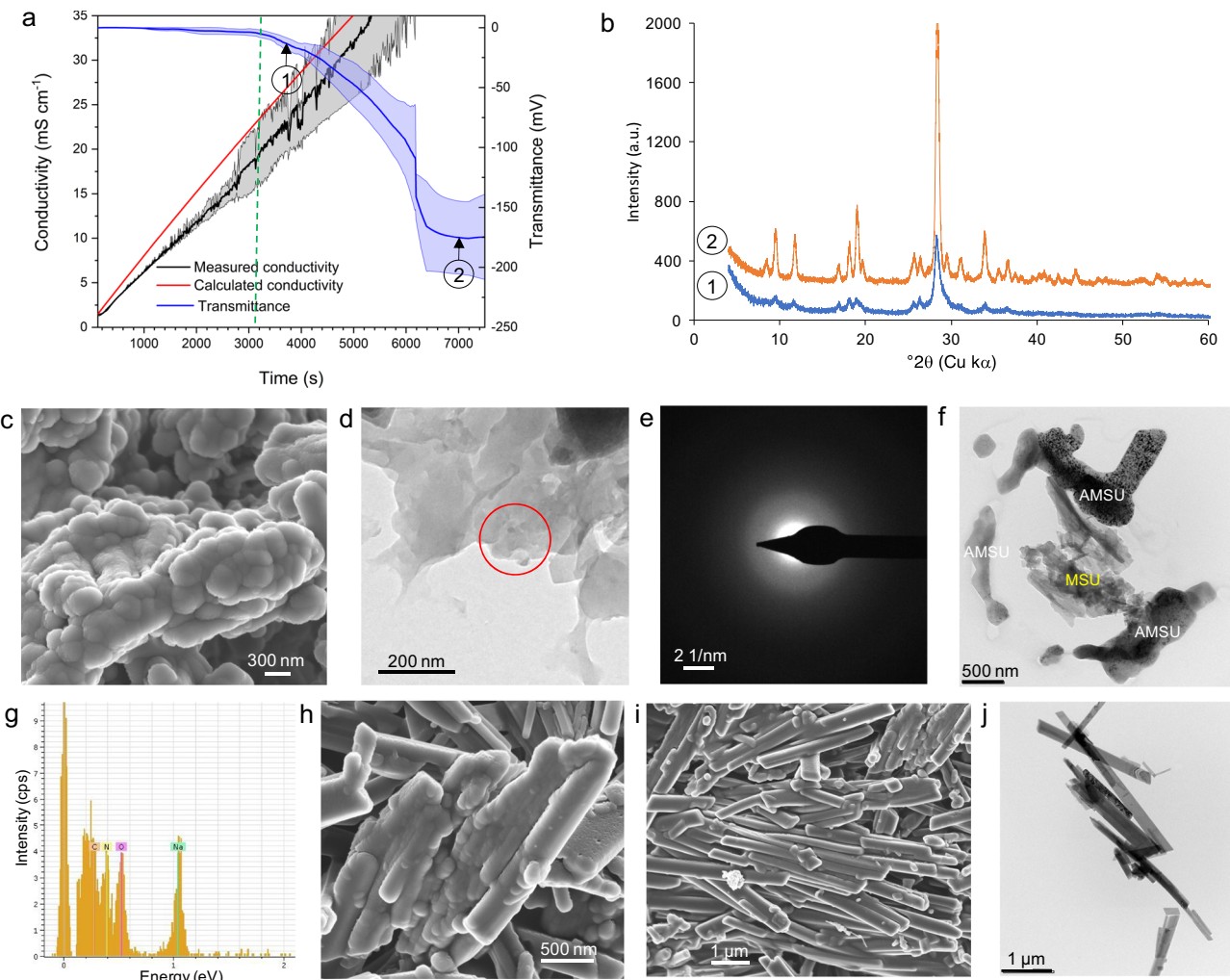

**Fig. 3 | Abiotic AMSU and its transformation into crystalline MSU. a** Time evolution of conductivity and transmittance during titration experiments resulting in the initial precipitation of AMSU (marked by the drop in transmittance, vertical blue dashed line). Shaded areas mark std dev. **b** XRD patterns of precipitates collected at the onset of precipitation (point 1 in (**a**)) and at the end of the titration test (point 2 in (**a**)). Note the marked increase in intensity and reduction in peak broadness in the latter case; **c** FESEM image of the nanogranular structure of early precipitates (AMSU); **d** TEM image of an aggregate of AMSU, as revealed by the diffuse haloes in the SAED pattern (**e**). **f** TEM image of the transformation of AMSU into MSU. **g** The EDS spectrum confirms that these phases are sodium urates. **h** FESEM image of an aggregate of poorly crystalline MSU showing a nanogranular surface structure; **i** FESEM and **j** TEM images of an aggregate of well-formed acicular (blade-like) MSU crystals. The latter (**i–j**) correspond to crystals formed in a silica gel, whereas the former (**c–h**) correspond to precipitates obtained in titration tests.

$K_{sp}$ is the solubility product of the relevant phase at 37 °C: $K_{sp}$ of MSU = $10^{-4.28}$ (ref. [46]), $K_{sp}$ of AMSU = $10^{-3.52}$, calculated considering that the solubility of AMSU is ~2.4 times higher than that of MSU[29]. $SI_{MSU}$ ranged from 1.61 to 1.72 ($SI_{AMSU}$ 0.85–0.96), corresponding to 170–220 mM [Na$^+$] and 12.5–12.3 mM [HU$^-$] at the onset of precipitation. In agreement with previous results[19], these very high $SI$ values show that under physiological pH and $T$, spontaneous MSU (or AMSU) precipitation (i.e., homogeneous nucleation) is very unlikely in hyperuricemic individuals.

**High-resolution synchrotron powder X-ray diffraction (HRXRD)**
To gain further insights into the possible formation mechanism of biotic MSU and the structural features a biomineralization process similar to that leading to the formation of other well-studied biominerals (e.g., carbonate shells) could imprint in MSU, we performed high-resolution synchrotron powder XRD (HRXRD) analysis of biotic and abiotic MSU. HRXRD analysis (Fig. 4) showed that all 0k0 and 0kl Bragg peaks of the biotic MSU crystals were right shifted (i.e., the d-spacing of the corresponding lattice planes were shifted to smaller values), whereas the h00 Bragg peaks were left-shifted (i.e., to higher d-spacings) as compared with the abiotic control.

Remarkably, biotic MSU displayed Bragg peaks with reduced broadening as compared with abiotic MSU. These effects were observed in biotic MSU crystals dispersed in SF as well as in tophi of different gouty patients (Fig. 4). Unit-cell refinement using a full-profile fitting approach (see Methods) showed that biotic MSU had a positive (expansion) lattice strain of $0.6 \times 10^{-3}$ ($\pm 1.9 \times 10^{-4}$) along the $a$-axis, and negative (shrinking) lattice strain of $-3.7 \times 10^{-3}$ ($\pm 1.8 \times 10^{-4}$) and $-2.1 \times 10^{-3}$ ($\pm 1.3 \times 10^{-4}$) along the $b$- and $c$-axis, respectively, as compared with abiotic MSU (Table 1). These lattice strain values show an overall lattice shrinkage (i.e., reduction of unit cell volume, Table 1) in the case of biotic MSU as compared with abiotic MSU (or lattice expansion of the latter as compared with the former, see below). Note that the Bragg peak positions and lattice parameters of the abiotic controls were in good agreement with those of the MSU structure proposed by Mandel and Mandel[36]. A possible reason for the observed lattice distortion (shrinkage) could be the incorporation in biotic MSU of cations present in SF with ionic radius smaller than that of Na$^+$, as it is the case of Mg$^{2+}$. However, our HAADF-EDS analyses showed no detectable amounts of magnesium in the biotic MSU crystals. We also considered that such lattice strain could be due to the occlusion of organic molecules within MSU crystals, as demonstrated in the case of several CaCO$_3$ biominerals[35,47].

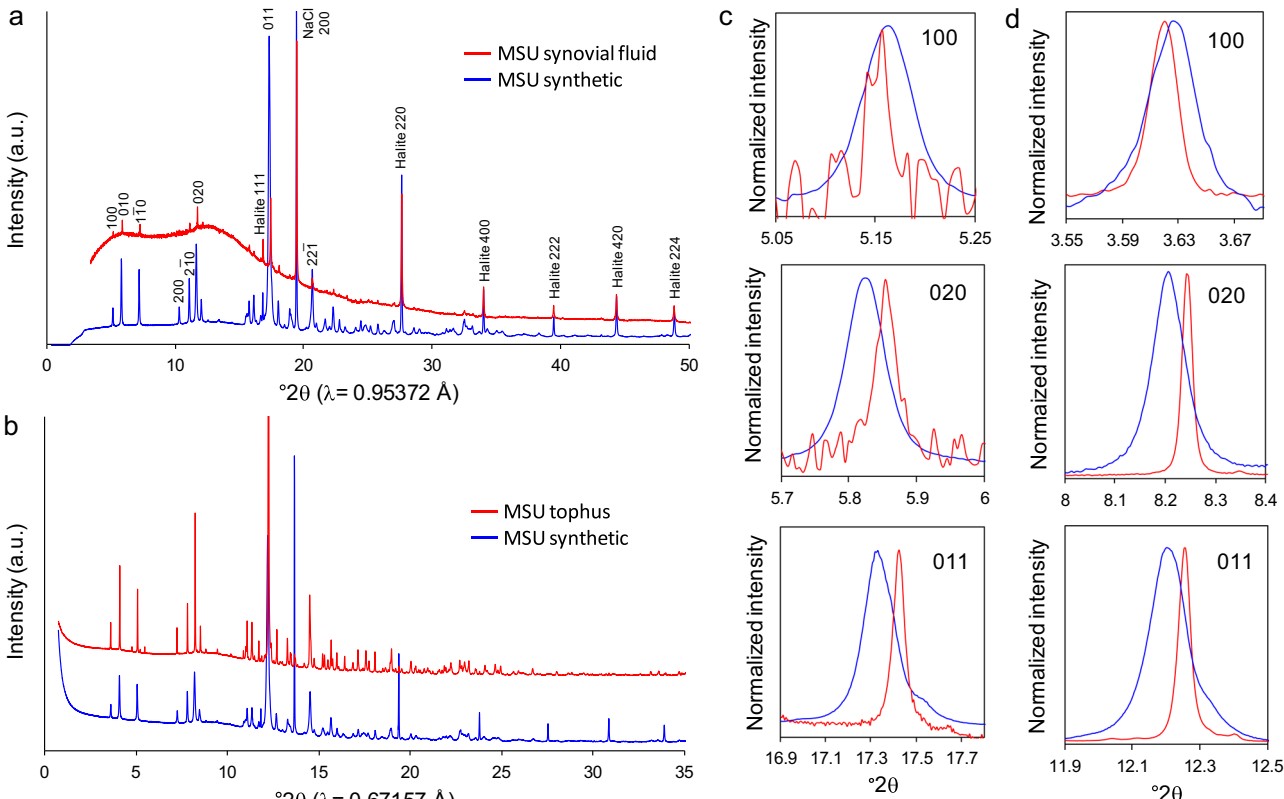

**Fig. 4 | Synchrotron HRXRD analysis of synthetic and biogenic MSU. a** Full HRXRD patterns of synthetic and biotic MSU (from synovial fluid of a gouty patient). Note the marked bump at 2θ < 25° due to the presence of amorphous organics in the synovial fluid. Also note the presence of halite (NaCl) in both samples, due to the high content of sodium chloride in synovial fluid and the use of NaCl as reactant in the synthesis of abiotic MSU. Data collected at ALBA synchrotron; **b** Full HRXRD patterns

of synthetic MSU and MSU collected from a human tophus. Data collected at SOLEIL synchrotron. Note that the X-ray wavelength in (**a**) and (**b**) is different; **c** detail of (**a**) showing the 100, 020 and 011 Bragg peaks of synthetic (blue line) and synovial-fluid derived MSU (red line); **d** Detail of (**b**) showing the 100, 020 and 011 Bragg peaks of synthetic (blue line) and tophus derived MSU (red line).

However, intracrystalline incorporation of organic macromolecules in such biominerals typically results in lattice expansion not in contraction. It also leads to a decrease in crystallite size and an increase in microstrain fluctuations (i.e., peak broadening)[47], which was not observed here. Indeed, it was very puzzling to observe that the crystallite size of biotic MSU was ~2.6 times larger than that of abiotic MSU, whereas the microstrain fluctuations of the former were ~2.6 times smaller than those of the latter (Table 1). These results show that biotic MSU includes a lower number of defects than the abiotic control and suggest that the structure reported by Mandel and Mandel[36] for abiotic MSU, precipitated in the laboratory under very high supersaturation conditions (Seegmiller's recipe)[5] similar to those of our abiotic MSU controls, reflects a strained structure with an intrinsic defect-related lattice expansion. Extensive research has demonstrated that there is a direct relation between defect (point, line and planar defects) density, Bragg peak broadening, and anisotropic lattice expansion in a range of ionic and molecular crystals[48–51]. A high defect density is typically observed when crystallization occurs at high supersaturation, and as a result, crystals grow at a fast rate[52,53]. Hence, biotic MSU displays the structure of highly perfect (defect-poor) MSU crystals, whereas the control MSU actually displays lattice expansion along *b* and *c* axes and contraction along the *a* axis due to its high defect density. This is a critical difference between biotic and abiotic MSU crystals that offers clues about their formation as we discuss below.

Altogether, the HRXRD results show that unlike other biominerals where organics occlusion is responsible for anisotropic lattice distortion[35,47], gouty MSU crystals display no apparent occlusion of organics and have a structure with higher perfection and crystallinity than that of abiotic MSU. This is a clear indication that biotic MSU forms under close-to-equilibrium conditions, that is, low supersaturation and very slow crystal growth rates,

factors that are key to the formation of crystals with a very low defect density. Calvert et al.[54] determined the growth rate of MSU crystal seeds in vitro, concluding that extrapolation of their results to actual physiological conditions in human SF with moderate (9 mg/dl) to severe (14 mg/dl) urate serum levels, imply that growth of average size pathological MSU crystals would take years. Note that Calvert et al.[54] fitted their growth rate model to a 2D island nucleation and spreading mechanism, which was demonstrated to be the actual growth mechanism of MSU by in situ AFM experiments[40]. Such a slow growth rate is consistent with the (micro)structural features of biotic MSU unveiled by our synchrotron HRXRD analysis. The opposite occurs in laboratory-synthesized MSU due to the very high supersaturation and fast growth rates. The presence of collagen fibrils favoring heterogeneous nucleation of a precursor AMSU, and its transformation into crystalline MSU at a relatively low supersaturation (marked by the solubility of AMSU, ~16.4 mg/dl)[29] appears to be key for the formation of phlogistic MSU with the observed nano and microstructural features in gouty patients.

## Discussion
By using different spectroscopic techniques, in combination with optical and electron microscopy, and synchrotron HRXRD, we observe distinctive features, such as the nanogranular surface texture and the high degree of crystalline perfection, that markedly differentiate biogenic MSU from its synthetic counterpart and demonstrate that biogenic MSU forms after a precursor AMSU, growing via a non-classical crystallization mechanism. Unlike classical crystallization, where a critical nucleus of the crystalline phase is formed after overcoming a high energy barrier, and further grows by incorporation of building units (ions or molecules), non-classical crystallization involves the formation of metastable precursors (ions associates,

**Table 1 | HRXRD derived lattice parameters, crystallite size (L) and microstrain fluctuations (ε) of biogenic and abiotic (synthetic[a]) MSU**

| MSU | a (Å) | b (Å) | c (Å) | α (°) | β (°) | γ (°) | Unit cell vol. (Å³) | L (nm) | ε |
|---|---|---|---|---|---|---|---|---|---|
| Biogenic tophus[b] | 10.8745(5) | 9.4800(9) | 3.5549(2) | 95.46(2) | 99.38(2) | 96.99(8) | 356.39(7) | 489.1 | 0.0011 |
| Synthetic I[b] | 10.8662(2) | 9.5171(8) | 3.5622(8) | 95.02(9) | 99.46(3) | 97.24(4) | 358.26(3) | 179.8 | 0.0043 |
| Biogenic SF[c] | 10.8755(2) | 9.4811(4) | 3.5563(5) | 95.45(1) | 99.34(2) | 96.97(7) | 356.68(6) | 362.4 | 0.0021 |
| Synthetic II[c] | 10.8701(2) | 9.5155(8) | 3.5643(8) | 95.01(3) | 99.53(4) | 97.25(1) | 358.48(6) | 150.3 | 0.0041 |
| Average lattice strain biogenic (Δx/x in %) | 0.06 ± 0.002 | −0.38 ± 0.02 | −0.22 ± 0.01 | | | | | | |

[a]Abiotic MSU crystals obtained in two different titration runs.
[b]HRXRD analyses performed at SOLEIL synchrotron.
[c]HRXRD analyses performed at ALBA synchrotron.

liquid and/or amorphous solid phases) after overcoming much lower energy barriers[34]. Following an amorphous-to-crystalline phase transformation, further growth takes place by ion/molecules incorporation and/or by attachment of nanoparticles[34,42–44]. In the case of biomineralization, non-classical crystallization enables an exquisite control on where and how a biomineral is formed and shaped after a precursor amorphous phase, typically with the aid of macromolecules[34,42–44], imprinting textural and (nano)structural features that enable to disclose its genesis.

The non-classical formation mechanism and specific textural/(nano) structural features of biotic MSU show that its crystallization is unlikely to be induced by a stereochemical (lock-and-key) interaction with specific proteins (e.g., albumin or IgG) as previously thought[20,26,27]. We observe the initial formation of biogenic AMSU associated with collagen fibrils in the SF (i.e., heterogeneous nucleation), and its transformation into MSU crystals with a nanogranular surface structure, likely associated with the interaction of the amorphous precursor nanoparticles with macromolecules in the SF, as also observed in $CaCO_3$ biominerals[34]. Moreover, we observe biotic micrometer-sized spherulitic aggregates of AMSU resembling the so-called MSU beachballs[39]. However, the latter, previously observed both in vivo[39] and in vitro[24,40], were made up of crystalline MSU. It is very likely that such crystalline spherulites are the result of the transformation of AMSU spherulites into MSU. This is in agreement with results by Li et al.[30] who observed in vitro the initial precipitation of spherical aggregates of AMSU, which eventually transformed into acicular MSU crystals. It follows that AMSU spherulites are a relevant reservoir for the subsequent formation of abundant phlogistic MSU crystals. We also observe minimal lattice distortion and microstrain fluctuations in biogenic MSU as compared with abiotic controls. Importantly, such features are present in biotic MSU regardless of whether crystals come from the SF or tophi in gouty patients, thus pointing to a common origin. These nano and microstructural features are fingerprints of the pathological biomineralization of MSU and help disclose how biogenic MSU forms and causes acute gout episodes.

We show that collagen fibrils enable heterogeneous nucleation of AMSU, and likely act as a template for the subsequent formation of MSU after AMSU, forming oriented MSU structures on fibrillated (damaged) collagen as those described by Pascual et al.[16]. Such structures are very similar to those observed in a range of biominerals[32]. In this respect, there is growing evidence showing a link between joint-damage related events associated with OA and gout pathogenesis[55,56]. Our results are in line with this evidence and point to OA and joint cartilage damage (fibrillation favoring heterogeneous nucleation of AMSU) as triggering factors for the development of gout. Hence, preventing/minimizing progression of OA and associated cartilage damage and fibrillation, given that these fibers clearly act as promoters for AMSU heterogeneous nucleation, could play a role in preventing the progression from asymptomatic hyperuricemia to gout.

Nonetheless, other factors likely contribute to the triggering of MSU crystallization and the development of gout[2,11]. Of note, is the reduced solubility of MSU with decreasing T (down to ~26 °C in the joints of lower extremities)[54], acidosis[11], -although this effect has been challenged[1]-, and/or the possible promoting effect (on AMSU nucleation) of different (macro)

molecules present in SF (e.g., proteins such as IgG and albumin, or carbohydrates such as chondroitin sulfate)[20,21,23,26,27], as well as small molecules such as acetate (a by-product of alcohol metabolism)[25] These factors likely act in synergy with damaged collagen fibers and should be further studied. Another aspect that should also be addressed is the effect of the localized production of uric acid (and the resulting increase in urate levels) in joint space associated with inflammation, cell death, and the development of OA, which likely plays a role in AMSU/MSU formation and gout development[57,58].

Finally, an important result of this study is the unexpected observation of the high perfection and crystallinity of biotic MSU, which implies very slow growth. This shows that it is very unlikely that gout flares are triggered by the sudden and massive crystallization of MSU in the joint cavity. Indeed, the presence of (typically scarce) MSU crystals has been demonstrated by microscopy in SF of joints never inflamed[59,60], and they are also observed in SF during intercritical periods[61], which agrees with the observation that for an inflammatory response (e.g., marked expression of CD86 on dendritic cells) a critical mass of MSU (≥1.0 mg/ml) is necessary[57]. MSU crystals could still be abundant following gout remission but rendered non-phlogistic due to a change in the protein coating, e.g., from IgG to Apo-B[62]. Our results indicate that prior to a gout flare, MSU crystals are slowly deposited and accumulated (e.g., in microtophi) on synovial tissue, and/or in aggregates developed on damaged joint cartilage (as well as in well-developed tophi) and are eventually shed to the joint cavity where they can trigger flares[1,2]. This provides a time-window to prevent the development of a critical mass of fully grown MSU crystals deposited on/within tissues surrounding joint cavities, that can trigger a gout flare when suddenly shed into the joint space. Yet what exactly triggers this event remains undefined[63]. Our results may thus aid in the development of novel treatments to prevent this illness by directly acting in the early stages of biomineralization of these pathological crystals. This could be achieved, for instance, by preventing the progression of cartilage damage in OA, and/or implementing urate lowering therapies once the very first AMSU (and/or MSU) precipitates are detected in joints of hyperuricemic subjects with subclinical inflammation using novel high resolution imaging techniques such as ultrasonography and dual energy CT[13], as well as detailed SF analysis. An early urate lowering therapy would be particularly effective for the dissolution of precursor AMSU due to its higher solubility as compared with MSU, thereby preventing its eventual transformation into phlogistic MSU.

## Materials and methods
### Collection of biotic MSU
SF aspirates were collected aseptically from 5 patients with gouty arthritis (two of them with tophi development and oligo-polyarticular flare). A large amount of MSU crystals were aspirated in the case of a patient showing a milky SF (so-called urate milk)[38]. In another case it was possible to collect a portion of a tophus with a surgical needle. All patients were hyperuricemic, with serum urate levels > 8 mg/dl. Synovial fluid samples were stored remnants collected from anonymous patients with crystal-related arthritis during standard clinical practice. The study was approved by the local ethics committee (CEIm, ref. PI2022-118). Informed consent was waived by the

ethics committee as there is no recording of personal information or identification along with the samples. Samples were stored at 5 °C in closed vials prior to analysis. Aliquots were centrifuged (10 min, 3000 rpm) to separate precipitates from SF. Concentrated solids were freeze-dried and stored in closed vials prior to analysis.

## Synthesis of abiotic MSU

MSU crystals were synthesized in the laboratory using three different routes. The first one involved the use of a silica gel as the porous medium for the diffusion-controlled crystallization of MSU following the procedure outlined in Parekh et al. [64]. After ca. 48 h, precipitation took place and MSU crystals were collected, dried under vacuum at room $T$, and stored in Eppendorf tubes for further analysis. Route two involved the rapid mixing in a reactor (under stirring) of uric acid (13 mM) and NaOH (0.1 M) + NaCl (140 mM) solutions at 37 °C, pH 7.4 ± 0.4 according to the method proposed by Perrin et al. [41]. Right after mixing a white precipitate formed. The solution was vacuum filtered, and the precipitates were dried in an oven at 40 °C. Once dried they were stored in Eppendorf tubes for further analysis. Route three involved the titration of 3 M NaCl solution into a 13.3 mM uric acid buffer whose pH was adjusted by adding 0.1 M KOH using an automatic burette (Dosino, Methrom). Note that KOH was selected as a base to neutralize uric acid as neutralization with NaOH resulted in the uncontrolled precipitation of MSU. The urate buffer (100 mL) was placed in a double-jacketed glass reactor at a constant $T$ of 37 °C (using a $T$-controlled water bath). $T$ and pH values were selected to mimic physiological conditions. To the urate buffer, 3 M NaCl solution was added at a constant rate of 0.01 mL per second using an automatic burette. pH, solution conductivity, temperature and transmittance were continuously recorded during the experiments using a Methrom Titrando equipment equipped with a glass electrode, a conductometric cell and a photometric sensor including a laser at a wavelength of 610 nm by Methrom. Aliquots of the solution with dispersed precipitates were collected at the onset of precipitation and once the precipitation was completed. Collected dispersions were immediately vacuum filtered and dried under vacuum at room $T$.

## Calculation of theoretical conductivity evolution during titration experiments

Theoretical electrical conductivity values ($\kappa_{cal}$) of the crystallization solution can be computed following the procedure outlined in our previous studies[44,65], assuming that all ions are free in solution. Here, we calculated the ionic molal conductivity ($\lambda$) for each of the ions (except for urate, see below) using equation:

$$\lambda_i = \lambda_i^\circ(T) - \frac{A(T)I^{\frac{1}{2}}}{1 + BI^{\frac{1}{2}}} \tag{1}$$

where $\lambda_i^\circ$ and $A$ are temperature ($T$, °C) dependent, $B$ is an empirical constant and $I$ is the ionic strength of the solution. The equations for $\lambda_i^\circ$ and $A$ calculation and $B$ values used here for the relevant ions in the system are those reported in ref. [66]. $I$ can be calculated as:

$$I = \frac{1}{2}\sum m_i z_i^2 \tag{2}$$

where $z_i$ is the charge of the $i$th ion. $\lambda_i^\circ$ and $A$ functions and $B$ value for the urate ion are not available in the literature; thus, we used the ionic molar conductivity (independent of the ionic strength) calculated from data reported in Mikulski et al. [67] (32.1 S cm² mol⁻¹).

Our calculations yield higher $\kappa_{cal}$ values than those measured ($\kappa$) as a result of ion clustering before nucleation:

$$\kappa = \kappa_{cal} - c_{Na-HU}\lambda_{Na-HU} \tag{3}$$

where $c_{Na-HU}$ is the concentration of pre-nucleation Na-HU associates and $\lambda_{Na-HU}$ is the molal conductivity of Na-HU associates.

Na activity measurements using selective ion electrodes were not reliable in our system under the experimental conditions used since they were not able to show any decrease in free Na⁺ activity upon solids formation, detected from changes in conductivity and turbidity measurements. Na⁺ and HU⁻ activities were thus determined using PHREEQC and the actual concentrations in solution (considering the dilution introduced by the constant addition of NaCl solution) as input. Note, however, that this calculation gives threshold (maximum) activity values, since ion association equilibrium constants of prenucleation associates are not included in the PHREEQC database (and are not known) and Na-HU association cannot thus be taken into account for activity calculations.

## TG/DSC, XRD, FTIR, Raman and FESEM analyses

Biotic and abiotic solids were subjected to simultaneous TG and DSC analysis on a Mettler-Toledo mod. TGA/DSC. About 10–20 mg sample mass was deposited on alumina crucibles and analyzed under flowing air (100 mL/min) at 10 °C min⁻¹ heating rate, from 25 to 950 °C. Additionally, solids were deposited on zero-background Si sample holders and analyzed on a Philips X'Pert Pro X-ray diffractometer equipped with Cu Kα radiation ($\lambda$ = 1.5405 Å), Ni filter, 3 and 70 °2θ range and at a scanning rate of 0.002 °2θ s⁻¹. Solids were also analyzed on a JASCO 6200 FTIR (frequency range 400–4000 cm⁻¹; 1 cm⁻¹ spectral resolution) equipped with an attenuated total reflectance (ATR) device for spectra collection without sample preparation (i.e., minimizing artifacts such as dehydration of MSU). Finally, solids were observed at high magnification using a FESEM (Zeiss Supra40VP or Zeiss Gemini). Samples were carbon coated prior to FESEM observation.

## TEM analysis

Analysis of the morphology, size, and ultrastructure of solids was performed by means of TEM using a FEI Titan, operated at 300 kV or a FEI Talos operated at 200 kV. Prior to TEM observations, solids were dispersed in ethanol, sonicated for 1 min and deposited on carbon/Formvar® film coated Cu grids. TEM observations were performed using a 30 μm objective aperture. SAED patterns were collected using a 10 μm aperture, which allowed collection of diffraction data from an area ~0.2 μm in diameter. EDS microanalysis was performed in scanning-TEM mode (STEM). Z-contrast imaging was performed using a HAADF detector (STEM mode).

## Synchrotron High Resolution Powder X-ray Diffraction analysis (HRXRD)

HRXRD analysis was performed at beamline BL04-MSPD of the ALBA synchrotron (Barcelona, Spain) and at beamline CRYSTAL of the SOLEIL synchrotron (Paris, France). The wavelengths 0.95372 Å (13 keV) and 0.67157 Å (18.462 keV) were selected for ALBA and SOLEIL, respectively, with a double-crystal Si (111) monochromator and determined by using Si640d NIST standard ($a$ = 5.43123 Å). Both ALBA and SOLEIL beamlines are equipped with a high-throughput position sensitive detector (PSD) MYTHEN optimal for time-resolved experiments. Borosilicate glass capillaries of 0.7 mm diameter were loaded with powder samples and rotated during data collection to improve diffracting particle statistics. The data acquisition time was ~5 min per pattern, with up to two iterations per measurement to obtain a good signal-to-noise ratio over the 3–95 and 2–55 °2θ angular range for ALBA and SOLEIL, respectively. To calibrate equipments and reduce possible shifts during experimental analysis, NAC standard (Na₂Al₂Ca₃F₁₄) and LaB₆ standards were used for ALBA and SOLEIL, respectively. An instrumental resolution factor analysis was performed in each case. Under these data acquisition conditions, the angular resolution was better than 0.006° FWHM.

The Rietveld refinement method[68] was used to extract lattice parameters of MSU matching the experimental diffraction peaks with those included in ref. [36] ($P\bar{1}$, $a$ = 1.0888 nm, $b$ = 0.9534 nm, $c$ = 0.3567 nm, $\alpha$ = 95.06°, $\beta$ = 99.47°, and $\gamma$ = 97.17°), using Topas 5.0 software.

To investigate the microstructural variations the Thompson-Cox-Hastings pseudo-Voigt function was used to fit the profile of the

experimental diffraction patterns of both biotic and abiotic crystals. Average crystallite size ($L$) and microstrain fluctuations ($\varepsilon$) were calculated using the Williamson-Hall plot method[69]. Fifty refinement cycles were performed to ensure reproducibility of each analysis, selecting a Goodness of Fit (GoF) of < 5. These parameters reduce the discrepancy between experimental and fitting values in a normal statistical distribution.

## Statistics and reproducibility

The sample size for human MSU samples was $N = 5$, whereas the sample size for synthetic MSU was $N = 50$. In the case of laboratory synthesized MSU, a minimum of three replicates were performed for each synthesis method to ensure reproducibility and statistical significance. Standard deviation for relevant data presented here shows 1σ values.

## Reporting summary

Further information on research design is available in the Nature Portfolio Reporting Summary linked to this article.

## Data availability

All data needed to evaluate the conclusions in the paper are present in the paper and/or the Supplementary Information. Raw data used to plot the different graphs in the main text and supplementary information figures are included in Supplementary Data 1–5 (Excel files). All other data are available from the corresponding author on reasonable request.

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

## Acknowledgements

This work has been funded by Spanish Government grant PID2021.125305NB.I00 funded by MCIN/ AEI /10.13039/501100011033 and by ERDF A way of making Europe; Junta de Andalucía and ERDF grant B-RNM-574-UGR20 and researchgroup RNM-179; University of Granada, Unidad Científica de Excelencia UCE-PP2016-05. K.E. is the recipient of grant RYC2022-037633-I financed by MCIN/AEI/10.13039/501100011033 and FSE+. We thank the personnel of the Centro de Instrumentación Científica (CIC) of the University of Granada for helping with FESEM and TEM analyses. HRXRD experiments were performed at BL-04 MSPD beamline at ALBA Synchrotron with the collaboration of ALBA staff (F. Fauth) and at CRYSTAL beamline at SOLEIL Synchrotron with the help of this beam line staff (E. Elkaim). We also thank P. Alvarez, M. Burgos-Ruiz, and J. Atienzar for help during HRXRD analyses at ALBA and SOLEIL, and C. Verdugo-Escamilla for his help with the Rielveld refinement. Finally, we thank L. Huber and S. Bonilla-Correa for their help with titration experiments.

## Author contributions

Conceptualization: C.R.-N. E. R.-A. and E.P.; Methodology: C.R.-N., K.E., and E.R.-A. Investigation: K.E., C.R.-N., L.M.-G., E.R.-A., A.I.-V., E.P., M.A., F.S.; Writing (original draft): C.R.-N.; Writing (review and editing): K.E., C.R.-N., L.M.-G., E.R.-A., A.I.-V., E.P., M.A., F.S.

## Competing interests

The authors declare no competing interests.
