## [Peer Review File · Communications Biology]

Reviewers' comments:

Reviewer #1 (Remarks to the Author):

In this study, the authors unveil a novel non-classical crystallization route, where MSU crystals form after a non-inflammatory amorphous precursor (AMSU) that nucleates heterogeneously on collagen fibrils from damaged articular cartilage of gout patients.

The authors identified the presence of an amorphous sodium urate phase, AMSU, attached to collagen fibers. They suggest that in patients' synovial fluid, this non-classical crystallization and growth mechanism relies on this metastable amorphous (AMSU) precursor of MSU.

In vitro, they were able to demonstrate the rapid conversion of AMSU into MSU under their precipitation conditions. Their biophysical analysis reveals that gouty MSU crystals display no apparent occlusion of organics and have a higher perfection and crystallinity structure than abiotic MSU. These significant observations suggest that biotic MSU forms under close-to-equilibrium conditions, that is, low supersaturation and very slow crystal growth rates, factors that are key to the formation of crystals with a very low defect density. This indicates that other features besides sudden crystallization may regulate gout flares in patients.

This study is not only interesting but also presents convincing data. It opens several questions related to the mechanisms possibly involving the factors that regulate the inflammatory properties of these crystals in humans, which are beyond the scope of the manuscript.

I appreciate the authors' thorough work and the insights it provides.
I have no additional comment on this manuscript.

Reviewer #2 (Remarks to the Author):

This study, which carefully traces the non-classical formation of MSU via amorphous precursors in gout patients, is novel and will be of interest to many. By combining various spectroscopic techniques with optical microscopy, electron microscopy, and synchrotron radiation HRXRD, the authors observed features that clearly distinguish biogenic MSU from synthetic MSU, such as nanoparticle-like surface morphology and high crystallinity. This has allowed us to successfully elucidate the crystallization mechanism of biogenic MSUs. Furthermore, the findings obtained in this study may be useful for the development of new treatments for the prevention of gout. For these reasons, I strongly recommend publication of this paper.

In addition, I would like to request the authors' response to the following two points:

- 1) In line 257, there is a statement "0k0, 00l and 0kl" but there is no figure for 00l in fig.4. The sentence needs to be corrected or a figure needs to be added.
- 2) The value of Unit cell vol. for Synthetic II in Table 1 is considerably larger than the others, but there is no description about it. A discussion of this point should be added.

Reviewer #3 (Remarks to the Author):

Thank you for giving me the opportunity to review the paper!!!
I have read the manuscript carefully and listed my personal comments.
If you're any questions, please let me know.

In this thesis, Carlos Rodriguez Navarro et al. report the pathological biomineralization of monosodium urate monohydrate (MSUM) crystals in gout patients. The authors note that the spherulite morphology of MSUM may be an evidence of a non-classical crystallization pathway for MSUM. This study seems noteworthy but it lacks the novelty and significance for publication in *Communications Biology*.

1. The author's survey of the research background to the reported content of the paper appears to be unsatisfactory, and the references cited appear to be overall dated and not sufficiently comprehensive.

2. Spherulites of MSUM in vivo and in vitro have been systematically studied and reported by scholars. Including but not limited to the following papers:

[1] J. J. Fiechtner and P. A. Simkin, *Adv. Exp. Med. Biol.*, 1980, 122, 141–143.

[2] J. J. Fiechtner and P. A. Simkin, *JAMA, J. Am. Med. Assoc.*, 1981, 245, 1533–1536.

[3] N. W. McGill, A. Hayes and P. A. Dieppe, *Scand. J. Rheumatol.*, 1992, 21, 215–219.

[4] M. H. Chih, H. L. Lee, T. Lee, *CrystEngComm* 2016, 18, 290–297.

[5] Y. Zhou, X. Feng, T. Wang, Y. Tian, X. Cui. *CrystEngComm*, 2021, 23, 1439

I did not find a very valuable and significant novel element in the results of this report, and also, this paper does not mention as well as compare the findings of this report with these results that have been reported.

3. Recent reports have demonstrated that hyaluronic acid significantly inhibits MSUM crystallization, this result is inconsistent with this report's claim that hyaluronic acid appears to have no effect on MSUM crystallization.

[1] Y. Liu, R. Cheng, C. Ou, X. Zhang and T. Fu, *Cryst. Growth Des.*, 2020, 20, 2842–2846.

[2] M. H. Chih, H. L. Lee, T. Lee, *CrystEngComm* 2016, 18, 290–297.

4. The manuscript in its current form is not up to the standards of *Communications Biology*, hence it is not acceptable for publication.

Response to referees

Hereby we present our point-by-point response to the three referees' comments and suggestions (our response in blue, new added text in blue-italics).

We want to thank the three referees for their insightful comments and suggestions that helped us to improve the overall quality of our manuscript.

Reviewer #1 (Remarks to the Author):

In this study, the authors unveil a novel non-classical crystallization route, where MSU crystals form after a non-inflammatory amorphous precursor (AMSU) that nucleates heterogeneously on collagen fibrils from damaged articular cartilage of gout patients.

The authors identified the presence of an amorphous sodium urate phase, AMSU, attached to collagen fibers. They suggest that in patients' synovial fluid, this non-classical crystallization and growth mechanism relies on this metastable amorphous (AMSU) precursor of MSU.

In vitro, they were able to demonstrate the rapid conversion of AMSU into MSU under their precipitation conditions. Their biophysical analysis reveals that gouty MSU crystals display no apparent occlusion of organics and have a higher perfection and crystallinity structure than abiotic MSU. These significant observations suggest that biotic MSU forms under close-to-equilibrium conditions, that is, low supersaturation and very slow crystal growth rates, factors that are key to the formation of crystals with a very low defect density. This indicates that other features besides sudden crystallization may regulate gout flares in patients.

This study is not only interesting but also presents convincing data. It opens several questions related to the mechanisms possibly involving the factors that regulate the inflammatory properties of these crystals in humans, which are beyond the scope of the manuscript.

I appreciate the authors' thorough work and the insights it provides.

I have no additional comment on this manuscript.

We appreciate the detailed reading and evaluation of our manuscript by this referee, and we thank him/her for the positive comments.

Reviewer #2 (Remarks to the Author):

This study, which carefully traces the non-classical formation of MSU via amorphous precursors in gout patients, is novel and will be of interest to many. By combining various spectroscopic techniques with optical microscopy, electron microscopy, and synchrotron radiation HRXRD, the authors observed features that clearly distinguish biogenic MSU from synthetic MSU, such as nanoparticle-like surface morphology and high crystallinity. This has allowed us to successfully elucidate the crystallization mechanism of biogenic MSUs. Furthermore, the findings obtained in this study may be useful for the development of new treatments for the prevention of gout. For these reasons, I strongly recommend publication of this paper.

We thank this referee for the positive evaluation of our manuscript and for calling our attention to the two points indicated below.

In addition, I would like to request the authors' response to the following two points:

1) In line 257, there is a statement "0k0, 00l and 0kl" but there is no figure for 00l in fig.4. The sentence needs to be corrected or a figure needs to be added.

In the main text we erroneously indicate 00l Bragg reflection: 00l has been eliminated in the revised version of the main text. Note that we did not selected the 00l Bragg peaks to show peak shifting because their intensity is very low (relative intensity $I/I_0 < 2\%$) and, in particular, the 001 Bragg peaks is masked by the 10-1 and 3-10 peaks (data from Mandel and Mandel, J. Am. Chem. Soc. 1976, 98, 8, 2319–2323).

2) The value of Unit cell vol. for Synthetic II in Table 1 is considerably larger than the others, but there is no description about it. A discussion of this point should be added.

We thank this referee for pointing out this issue. We re-analyzed (full profile fitting and unit cell refinement using the Rietveld method) all HRXRD patterns of synthetic and biotic MSU. The re-analysis showed that unit cell parameters for all samples except for the Synthetic II matched the values reported in Table I. In fact, the new obtained value of the a parameter of Synthetic II was larger and the b parameter was smaller than that formerly reported in Table 1. As a result, the newly obtained unit cell volume was nearly identical to that of the Synthetic I sample. These results showed that the previously reported unit cell values for Synthetic II were not correct. In Table I of the revised version of the manuscript we now report the correct values, as we also do in the main text.

Reviewer #3 (Remarks to the Author):

Thank you for giving me the opportunity to review the paper!!!

I have read the manuscript carefully and listed my personal comments.

If you're any questions, please let me know.

In this thesis, Carlos Rodriguez Navarro et al. report the pathological biomineralization of monosodium urate monohydrate (MSUM) crystals in gout patients. The authors note that the spherulite morphology of MSUM may be an evidence of a non-classical crystallization pathway for MSUM. This study seems noteworthy but it lacks the novelty and significance for publication in Communications Biology.

We thank Dr. Liu for the thorough review of our manuscript.

We note that we present multiple lines of evidence indicating that the in vivo formation of MSU follows a non-classical crystallization pathway involving an amorphous precursor phase (AMSU). The presence of AMSU spherulites is just one example of this evidence. This is, to our knowledge, the first study that demonstrates that biotic MSU forms in humans via an AMSU phase. We strongly believe, in agreement with the comments by referees #1 and #2, that this study is novel and significant.

Below we present our response to each specific comment by this referee.

1. The author's survey of the research background to the reported content of the paper appears to be unsatisfactory, and the references cited appear to be overall dated and not sufficiently comprehensive.

We presented a balanced revision of relevant publications on the studied topic. Note that the aim of the study was not to present a comprehensive and systematic review of published papers on MSU formation. However, we acknowledge that we missed a few relevant papers, pointed out by Dr. Liu, which are now cited in the revised version of the manuscript (see below).

2. Spherulites of MSUM in vivo and in vitro have been systematically studied and reported by scholars. Including but not limited to the following papers:

[1] J. J. Fiechtner and P. A. Simkin, *Adv. Exp. Med. Biol.*, 1980, 122, 141–143.

[2] J. J. Fiechtner and P. A. Simkin, *JAMA, J. Am. Med. Assoc.*, 1981, 245, 1533–1536.

[3] N. W. McGill, A. Hayes and P. A. Dieppe, *Scand. J. Rheumatol.*, 1992, 21, 215–219.

[4] M. H. Chih, H. L. Lee, T. Lee, *CrystEngComm* 2016, 18, 290–297.

[5] Y. Zhou, X. Feng, T. Wang, Y. Tian, X. Cui. *CrystEngComm*, 2021, 23, 1439

I did not find a very valuable and significant novel element in the results of this report, and also, this paper does not mention as well as compare the findings of this report with these results that have been reported.

We fully agree with Dr. Liu statement. Indeed, there are many papers reporting the formation of MSU spherulites. We actually cited the seminal paper by Fiechtner and Simkin (1981) (our original ref. 37 -now ref. 39-, indicated by Dr. Liu in the above list), who were the first to observe the in vivo formation of MSU spherulites (they called them "beachballs") and also were the first to reproduce them in vitro. In any case, for completeness, in the revised version of the Ms we now cite the papers by Chih et al. (2016) and Zhou et al (2021) indicated by Dr. Liu in the list above. Note however that we do not cite the paper by McGill et al. (1992), because they do not report the formation of "beachball" or spherulitic MSU structures: they report the in vivo and in vitro formation of bow-shaped MSU aggregates.

Note that the observation of MSU spherulites in the references above refer to crystalline MSU, while we report the in vivo formation of amorphous MSU spherulites. This is a novel result, and we now contrast this finding with reported in vivo and in vitro formation of MSU spherulites. In the Discussion of the revised version of the manuscript we state: "*Moreover, we observe biotic micrometer-sized spherulitic aggregates of AMSU resembling the so-called MSU "beachballs"³⁹. However, the latter, previously observed both in vivo³⁹ and in vitro^{24,40}, were made up of crystalline MSU. It is very likely that such crystalline spherulites are the result of the transformation of AMSU spherulites into MSU. This is in agreement with results by Li et al.³⁰ who observed in vitro the initial precipitation of spherical aggregates of AMSU which eventually transformed into acicular MSU crystals. It follows that AMSU spherulites are a relevant reservoir for the subsequent formation of abundant phlogistic MSU crystals.*".

3. Recent reports have demonstrated that hyaluronic acid significantly inhibits MSUM crystallization, this result is inconsistent with this report's claim that hyaluronic acid appears to have no effect on MSUM crystallization.

[1] Y. Liu, R. Cheng, C. Ou, X. Zhang and T. Fu, *Cryst. Growth Des.*, 2020, 20, 2842–2846.

[2] M. H. Chih, H. L. Lee, T. Lee, *CrystEngComm* 2016, 18, 290–297.

We thank this referee for pointing out this fact. We now refer to the role of hyaluronic acid as a MSU crystallization inhibitor and cite the papers by Chih et al. (2016) and Liu et al (2020). Moreover, in the discussion, we also cite the paper by Liu et al. (2020) in the context of the role of acetate (a by-product of alcohol metabolism) as a promoter of MSU crystallization.

4. The manuscript in its current form is not up to the standards of *Communications Biology*, hence it is not acceptable for publication.

We strongly believe, in agreement with the comments by referees #1 and #2, that the paper is novel and relevant, and thus merits publication in *Communications Biology*.

REVIEWERS' COMMENTS:

Reviewer #2 (Remarks to the Author):

I mention below the two points I raised.

- 1) I confirmed that the noted 00l diffraction has been corrected in the text.
- 2) Regarding the point made in Table I about the unit cell volume values of Synthetic II being considerably larger than the others, the author has reanalyzed and corrected them to the correct values, which are almost the same as those of Synthetic I.

I thank the authors for their diligent response to these remarks.

Reviewer #3 (Remarks to the Author):

In the revised version, my main comments have been considered and addressed. Therefore, my personal opinion is that the paper is suitable to be recommended for publication.

Reviewer #4 (Remarks to the Author):

This is a highly significant and elegant study on gout, a highly prevalent disorder whose incidence is increasing globally. In this context, approximately 5% of USA adults have gout, yet asymptomatic hyperuricemia has a prevalence of over 20%. Moreover, advanced imaging studies that included use of ultrasound have revealed that at least a quarter of subjects with hyperuricemia have urate crystal deposition in joint tissues, with particular predilection for articular cartilage surfaces. Urate spherulites and de novo formation of monosodium urate (MSU) crystals in joint fluids are well enough described in the literature, and de novo formation and release from cartilage of MSU crystals likely account for a major driving force of gouty joint inflammation. This paper contributes translationally to gout by linking cartilage biology to MSU crystallization, and by identifying a novel and non-classical MSU crystallization pathway involving an amorphous precursor spherulite phase (AMSU) *in vivo*. Importantly, the precursor crystal phase is known to be non-inflammatory, suggesting a specific and interestingly, very prolonged target phase for intervention in hyperuricemia and gout, particularly in those with articular damage. And osteoarthritis and joint trauma are strongly linked to gout and gout flares, respectively. The data shown are thorough and convincing, and technically sophisticated. I very strongly support the significance and impact of this study.

Response to referees

Hereby we present our point-by-point response to the three referees' comments (our response in blue, new added text in blue-italics).

We want to thank the three referees for their insightful and encouraging comments.

Reviewer #2 (Remarks to the Author):

I mention below the two points I raised.

- 1) I confirmed that the noted 00l diffraction has been corrected in the text.
- 2) Regarding the point made in Table I about the unit cell volume values of Synthetic II being considerably larger than the others, the author has reanalyzed and corrected them to the correct values, which are almost the same as those of Synthetic I.

I thank the authors for their diligent response to these remarks.

We thank this referee for his/her thorough review of our manuscript and the final comments.

Reviewer #3 (Remarks to the Author):

In the revised version, my main comments have been considered and addressed. Therefore, my personal opinion is that the paper is suitable to be recommended for publication.

We thank this referee for his/her thorough review of our manuscript and the final recommendation for publication.

Reviewer #4 (Remarks to the Author):

This is a highly significant and elegant study on gout, a highly prevalent disorder whose incidence is increasing globally. In this context, approximately 5% of USA adults have gout, yet asymptomatic hyperuricemia has a prevalence of over 20%. Moreover, advanced imaging studies that included use of ultrasound have revealed that at least a quarter of subjects with hyperuricemia have urate crystal deposition in joint tissues, with particular predilection for articular cartilage surfaces. Urate spherulites and de novo formation of monosodium urate (MSU) crystals in joint fluids are well enough described in the literature, and de novo formation and release from cartilage of MSU crystals likely account for a major driving force of gouty joint inflammation. This paper contributes translationally to gout by linking cartilage biology to MSU crystallization, and by identifying a novel and non-classical MSU crystallization pathway involving an amorphous precursor spherulite phase (AMSU) in vivo. Importantly, the precursor crystal phase is known to be non-inflammatory, suggesting a specific and interestingly, very prolonged target phase for intervention in hyperuricemia and gout, particularly in those with articular damage. And osteoarthritis and joint trauma are strongly linked to gout and gout flares, respectively. The data shown are thorough and convincing, and technically sophisticated. I very strongly support the significance and impact of this study.

We thank this referee for his/her thorough review of our manuscript and the very positive and encouraging comments.